# Endoscopic Treatment of Upper Tract Urothelial Carcinoma: Challenging the Definition of the Maximal Lesion Size for Safe Ablation

**Yazeed Barghouthy** [1] , **Mariela Corrales** [1], **Alba Sierra** [1] , **Hatem Kamkoum** [2], **Camilla Capretti** [3],
**Bhaskar Somani** [4] , **Eva Compérat** [5,6] **and Olivier Traxer** [1,7,*]

1  Groupe de Recherche Clinique sur la Lithiase Urinaire, Sorbonne Université, GRC n° 20, Hôpital Tenon, 75020 Paris, France; yazeedmail@gmail.com (Y.B.); mariela_corrales_a@hotmail.com (M.C.); asierradelrio@gmail.com (A.S.)
2  Urology Department, Hamad Medical Corporation, Doha 3050, Qatar; hatemkamkoum@gmail.com
3  Institute of Urology, Polytechnic University of Marche, 60121 Ancona, Italy; Camilla.capretti2@gmail.com
4  University Hospital Southampton NHS Trust, Southampton SO16 6YD, UK; bhaskarsomani@yahoo.com
5  Department of Pathology, Hôpital Tenon, 75020 Paris, France; eva.comperat@aphp.fr
6  Favulty of Medicine, Sorbonne Université, 75006 Paris, France
7  Service d'Urologie, AP-HP, Hôpital Tenon, 75020 Paris, France
*  Correspondence: olivier.traxer@aphp.fr; Tel.: +33-1-56-01-61-53; Fax: +33-1-56-01-63-77

**Abstract: Introduction:** With advances in endoscopic treatment of upper tract urothelial carcinoma (UTUC) lesions, the recommended upper limit of lesion size amenable to laser ablation was set to 2 cm. However, this limit is based on expert opinion only, and debate still exists regarding this definition. Objective: To determine the maximal size of the tissue, for which total endoscopic ablation with laser energy is achievable, from a laser performance perspective. **Materials and Methods:** Simulating endoscopic surgery conditions, renal tissue blocks from pork kidneys in growing size from 1 cm$^3$ to 3 cm$^3$ were totally ablated with Ho:YAG laser (1 J, 10 Hz). The time to ablation was recorded for each tissue mass. Following the ablation, each sample was inspected microscopically by an expert pathologist to determine the extent to which the tissue was destroyed. **Results:** Time to ablation ranged from 16.4 min for a 1 cm$^3$ mass, to 69.7 min for a 3 cm$^3$ mass. Histologic evaluation after laser ablation showed that ablation was achieved in all tissue masses, and no "unaffected" tissue was present, even for lesions with a size of 3 cm$^3$. **Conclusions:** This study showed that laser ablation can be achieved for tumor lesions up to a size of 3 cm$^3$. The results of this study can contribute to the debate regarding the limits of endoscopic management of UTUC lesions and strengthen the recommended upper limit of 2 cm$^3$ for endoscopic treatment of tumor lesions.

**Keywords:** flexible ureteroscopy; upper tract; urothelial cancer; laser; lesion; size

## 1. Introduction

Upper tract urothelial cancer (UTUC) is a relatively uncommon malignancy (around 1–2 cases per 100,000), representing approximately 5–10% of all urothelial cancers (UC) [1]. Despite the common histologic features with bladder cancer (BC), UTUC tends to have a worse prognosis, mainly due to the higher percentage of invasive disease at diagnosis in around 60% of cases, compared to 20% in BC [2–4].

In the last two decades, with the significant improvement of endoscopic and imaging techniques, accompanied by a growing number of published studies [5–9], the management of UTUC has shifted towards a risk-stratified approach [10,11]. This allows many UTUC cases with a low risk of evolution to be treated through a nephron sparing approach, mainly endoscopic laser ablation of these lesions, with comparable results to radical surgery for low-risk disease, under the conditions of strict surveillance [12–14]. The mainstay of endoscopic treatment is laser ablation with Holmium:YAG (Ho:YAG) laser, which produces

tissue ablation reaching an incision depth (ID) of 2 mm and coagulation thickness of 0.48 mm [15]. Other lasers were also evaluated, and while the Thulium:YAG (Tm:YAG) laser was found to produce larger coagulation areas [16], the new super pulsed Thulium Fiber laser produced a smaller coagulation area and incision depths than the Ho:YAG laser [17].

Despite the accumulating experience over the last three decades, there is a lack of level 1 evidence regarding endoscopic treatment of UTUC due to the rare nature of this disease and the single institutional retrospective character of most of the available literature. One of the points of debate is the lesion size limit, above which endoscopic treatment is not recommended. While different cut-offs have been presented, ranging from 1 to 2 cm [18–21], these recommendations are based on expert opinions, and not evidenced from experimental studies [8].

For this reason, we performed this study to optimally evaluate, on a histological level, the size limit of lesions which can be managed with endoscopic laser ablation.

## 2. Methods

After acquiring the necessary approvals, a laboratory study was planned to simulate the ablation of urothelial tissue endoscopically by laser energy. Ten fresh non-frozen porcine kidneys were used as an ex vivo study model to evaluate laser-tissue ablation. Although various tissue models were used in the literature to investigate the effects of lasers on histological parameters, porcine kidneys were the most reported [15–17,22,23]. The kidneys were coronally sectioned and dissected. A total of Fifty tissue blocks of porcine kidney tissue were obtained and divided into 5 groups of growing volume sizes of: 1, 1.5, 2, 2.5 and 3 cm$^3$, forming groups 1, 2, 3, 4 and 5, respectively. A preliminary pilot study with 10 additional specimens was also performed.

The tissue blocks were placed in a 'renal pelvi-calyceal' collecting system model, filled with saline solution (Figure 1A–D). They were subsequently destroyed by laser energy, imitating the "real-life" endoscopic surgery, by using a 272 μm laser fiber connected to a 35 W Ho: YAG laser (Dornier Medilas H Solvo 35, Hamburg, Germany). The settings used were 1 J and 10 Hz as recommended by experts in the literature [24]. We used a single use flexible ureteroscope (*Lithovue®-Boston Scientific*) in order to visualize the tissue and perform the ablation endoscopically. The tissue ablation continued until no viable tissue was recognized endoscopically, by two experienced endourologists (YB, MC). The time to completion (in minutes) was measured, with an upper limit of 90 min for the ablation of each tissue block.

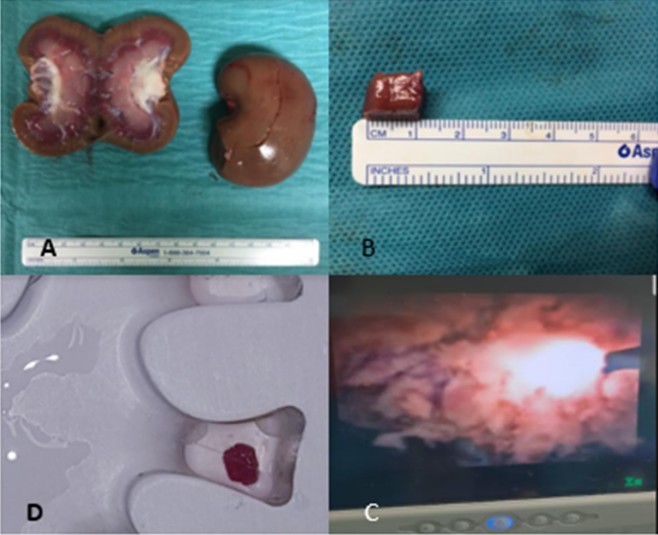

**Figure 1.** Experiment setup showing (**A**) Porcine kidney and section (**B**) 1 cm specimen of tissue (**C**) tissue block in collecting system model (**D**) Endoscopic view during tissue laser ablation.

Upon completion of this task, the tissue block and all available fragments were collected and fixated in a 4% formalin solution. Thereafter the tissue was processed, and H&E staining was performed. Histopathologic evaluation was performed by an expert pathologist in urogenital cancers (EC), and three categories were searched for: "non affected" tissue (representing "viable" non destroyed tissue in a real-life scenario), necrotic destroyed tissue but with recognizable architecture, or fragmented/destroyed (scorched) tissue by electro-coagulation, with an estimation of the percentage of tissue in each category.

For statistical analysis, we used SPSS Statistics 23.0 (IBM, Armonk, NY, USA). Group's time for complete ablation is expressed as mean—SD (range). Analysis of variances (ANOVA) was used to calculate the mean time for complete tissue ablation between groups. Statistical significance was set at $p < 0.05$.

### 3. Results

The mean tissue ablation time in minutes (min), presented in Table 1, ranged from 16.4 min for a 1 cm$^3$ tissue block to 69.7 min for a 3 cm$^3$ tissue block ($p < 0.05$). Figure 2 shows the different tissue forms under microscopic view, from normal tissue before ablation (A) to destroyed unrecognizable tissue (D).

**Table 1.** Results of ablation time and histological analysis of ablated tissue according to tissue size.

| Group | Time (min) to Complete Ablation. Mean (SD) | Percentage of Non-Affected Tissue | Percentage of Ablated Necrotic Tissue with Recognizable Architecture | Percentage of Destroyed Unrecognized Tissue |
|---|---|---|---|---|
| 1 (1 cm$^3$) | 16.4 (2.7) | 0% | 0% | 100% |
| 2 (1.5 cm$^3$) | 28.1 (2.5) | 0% | 0% | 100% |
| 3 (2 cm$^3$) | 40.3 (3.3) | 0% | 0% | 100% |
| 4 (2.5 cm$^3$) | 53.7 (4.3) | 0% | 8% | 92% |
| 5 (3 cm$^3$) | 69.7 (6.6) | 0% | 15% | 85% |

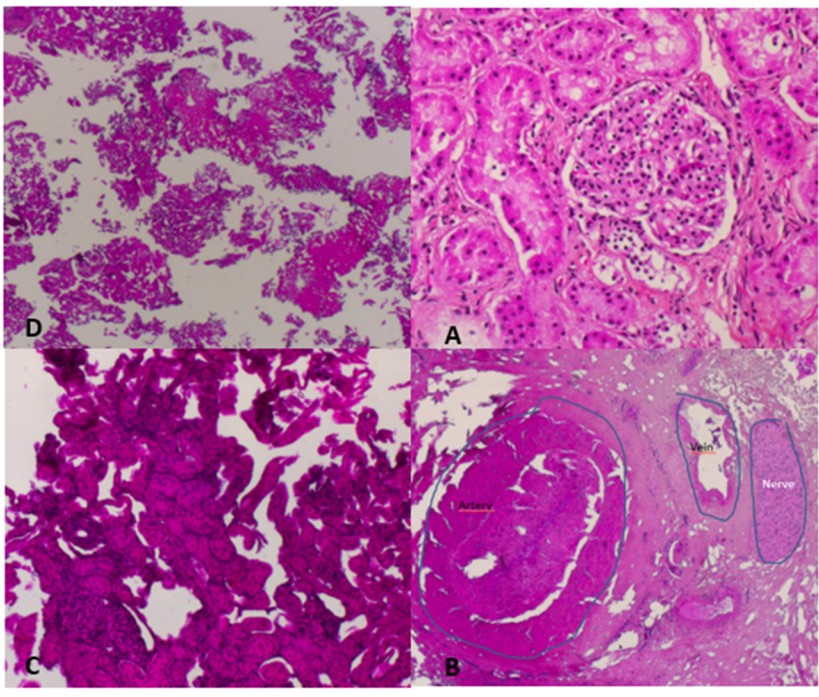

**Figure 2.** Clockwise: Microscopic illustration of (**A**) Normal Tissue (**B**) After laser Ablation showing necrotic tissue with recognized architecture (**C**) Ablated tissue and (**D**) Fragmented tissue.

Histologically, no unaffected tissue (representing "viable" tissue) was detected in any specimen (Figure 2). There were only two categories demonstrated microscopically, the first was necrotic tissue with recognizable tissue architecture to a certain extent, and the second was fully ablated (scorched) nonrecognizable tissue (Table 1).

## 4. Discussion

UTUC is a neoplasm arising from the urothelial lining anywhere in the upper urinary tract, with two-thirds of the cases presenting in the renal pelvis, and around one-third in the ureter. As mentioned previously, UTUC can present in an invasive stage in approximately 60% of cases [2–4]. Other features of UTUC are multifocality in about 33% of cases, and the metachronous appearance of lesions which can occur months to years after the diagnosis of the index lesion. These latter features are thought to be caused by a field change effect in the urothelium and by intraluminal seeding of tumor cells in the renal collecting system [24,25].

Research in the last 3 decades has allowed the recognition of risk factors for tumor evolution [26–32], thus permitting the stratification of tumors into low and high-risk groups. Consequently, a risk-oriented approach has been adopted to guide treatment, driven principally by the necessity to avoid a loss of renal function caused by radical surgery.

Accordingly, radical nephroureterectomy with complete distal ureteral resection including a bladder cuff is considered the gold standard for cases deemed to be high risk. While kidney preserving procedures (KPP), such as endoscopic management (ureteroscopic or percutaneous approach), and segmental ureterectomy could be considered for low-risk cases [5–8], the most widely used modality for kidney preservation is the endoscopic retrograde ureteroscopic approach, using laser energy for tissue ablation.

Ho:YAG laser is the principal energy source used, with the settings usually being 1 J for energy and 10 Hz for frequency, with a tissue penetration depth of 0.4 mm. Certain teams suggest also the use of Nd:YAG laser in addition to Ho:YAG due to its' deeper penetration in the tissue of around 5–6 mm [25,33].

The updated EAU guidelines recommend conservative management of UTUC for unifocal, low grade tumors, less than 2 cm in size, with no evidence of infiltration on computed tomography. Patients must also adhere to a strict protocol of follow-up, starting with a second look procedure 6 weeks after the primary tumor ablation [5]. Furthermore, imperative indications for endoscopic management of UTUC are a solitary kidney, bilateral tumors, significant medical comorbidities, inability to tolerate radical surgery and chronic kidney disease.

Tumor size itself is not a component of the TNM classification system in UTUC. However, it is taken into consideration when classifying the tumor into the low or high-risk categories, thus guiding the management choices. Previous recommendations have defined a limit of 1 cm, above which the tumor is considered high risk and endoscopic treatment is not recommended [20,21]. This limit has been changed to 2 cm in the updated recommendations [5]. Nevertheless, as mentioned previously, this limit has been based solely on expert opinions and a few retrospective case series [34–37].

Pieras et al. suggested that lesion size > 4 cm was associated with a significant rise in bladder recurrence rates in univariate and multivariate analysis [34]. While Simone et al. suggested that patients with tumors > 3 cm might have lower metastasis free survival than patients with smaller tumors [35].

Villa et al. evaluated the effects of tumor size, tumor distribution and tumor grade on progression-free survival (PFS) in patients with UTUC treated with flexible ureteroscopy (fURS) and Ho:YAG laser photoablation. At a median follow-up of 52 months, PFS rates were 68% vs. 72% in patients with tumor size < 1 cm vs. > 1 cm, respectively (*p* = 0.9). The study concluded that tumor size > 1 cm does not increase the risk of disease progression in patients treated endoscopically with fURS and laser ablation [36].

Scotland et al. published their experience with the fURS and laser ablation in 80 patients, with biopsy proven low grade UTUC, with at least one lesion larger than 2 cm (mean tumor size 3.04 cm). During a median follow-up of 43.6 months, 90.5% of tumors

had an ipsilateral recurrence, and 31.7% progressed in grade at a median follow-up of 26.3 months. RNU was performed in 16 patients (20%). The overall survival was 75% and cancer specific survival was 84% at 5-year follow up. The authors concluded that ureteroscopic management of large ($\geq$2 cm) UTUC lesions is a possible alternative to RNU, under the condition of compliance with strict surveillance [37].

The results of the current study showed that laser photoablation used in endoscopic treatment of UTUC is capable of entirely ablating a tissue mass as large as 3 cm$^3$, as demonstrated histologically.

Certain tissue blocks were not fragmented and kept a central tissue core intact. This raised the suspicion of residual tissue areas in the center-unaffected by the thermoablative effect of the laser- corresponding to "viable" tissue in real-life lesions. However, the histologic evaluation of these remnant blocks demonstrated the destruction of tissue in the center of the tissue block, as well as on the surface. This implies that no viable tissue could withstand endoscopic laser ablation of a tissue mass of up to 3 cm$^3$ (the largest tissue block we evaluated) and that the destruction of tissue takes place beyond our endoscopic view of the ablated surface. Noteworthy is that necrotic areas were continuous under microscopic view and not patchy, signifying that this necrosis was indeed the effect of laser on the entirety of the tissue block. One interesting finding was that areas of tissue with blood vessels were more resistant to destruction, probably due to the presence of fibrotic tissue in the vascular wall.

The authors measured tissue size in cm$^3$ and not cm units, to outline the fact that volume is more indicative of the ability to ablate a mass than its unidimensional size in cm. The importance of using volume in cm$^3$ instead of size has also been highlighted in urinary stones management [38,39].

This study has some limitations. The first is without doubt the use of porcine kidney tissue, different in its' histological characteristics than urothelial tumor tissue. However, the conditions in which the experiments were performed, are probably the closest available for the simulation of urothelial tumor ablation, as it is practically impossible to get real tumor tissue on which to run the experiment. Regarding histology, urothelial carcinoma can exhibit a wide range of variant morphologies. Non-invasive neoplasms share a similar morphological spectrum of intra-urothelial changes, ranging from hyperplasia to dysplasia to carcinoma in situ. The main difference from the surrounding non-neoplastic tissues is the architectural growth patterns of the tumor tissue [40]. The second limitation is that the experiments were not performed on live animals with perfused tissues, reproducing the surgical conditions, allowing us to assess hemostatic proprieties [41], and to achieve similar tissue composition, density and temperature compared to in vivo conditions. Therefore, our results may have been affected by this limitation. However, the Ho:YAG has proved to be a versatile laser in urology [15,16,22] demonstrating its' incision and coagulation capacities in similar tissue conditions existing in this study. This study used Ho:YAG laser only. However, based on previous results of soft tissue ablation with Thulium:YAG and Thulium Fiber Laser (TFL) [16,17,22,23,42], the authors believe that the same results can also be applicable with the use of TFL. Further ex vivo investigations could be performed with TFL.

The results of this study can contribute to the debate regarding the maximal tissue size that may be treated endoscopically. Nonetheless, the surgeon's experience and judgment, together with individual patient risk stratification, remain the most important factors in making the decision for each patient.

## 5. Conclusions

Concerning the capacity of lasers to ablate upper tract urothelial tumor masses, lesions up to 3 cm$^3$ are amenable to endoscopic treatment. This study strengthens the new definition of a maximum size of tumor lesions that can be suitable for endoscopic management.

**Author Contributions:** Y.B.: Project development, Data Collection, Manuscript writing. M.C., A.S., H.K. and C.C.: Data collection and analysis. B.S., E.C. and O.T.: Project development, Manuscript editing. All authors have read and agreed to the published version of the manuscript.

**Funding:** This is an independent study and is not funded by any external body.

**Institutional Review Board Statement:** Not applicable.

**Informed Consent Statement:** Not applicable.

**Data Availability Statement:** Not applicable.

**Conflicts of Interest:** The authors declare no conflict of interest. This study involved animal kidney tissue.

**Disclosure:** Olivier Traxer is a consultant for Coloplast, Rocamed, Olympus, EMS, Boston Scientific and IPG.

## Abbreviations

| | |
|---|---|
| UTUC | Upper Tract Urothelial Carcinoma |
| UC | Urothelial Carcinoma |
| BC | Bladder Cancer |
| KPP | Kidney Preserving Procedures |
| PFS | Progression free survival |
| fURS | Flexible Ureteroscopy |
| Ho:YAG | Holmium:YAG |

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
