# Peer review of "Endoscopic Treatment of Upper Tract Urothelial Carcinoma: Challenging the Definition of the Maximal Lesion Size for Safe Ablation"

_2673-4397, doi:10.3390/uro2010003_

Round 1

Reviewer 1 Report

This is an interesting paper presenting results in a non-human model.

The aim of the study is to determine the minimal size of tissue that could be ablated with a laser energy during an endourologic surgery.

I would suggest to better specify how tissue block are prepared in a collecting system model and how was simulated the endourological treatments.

I would also add in the methods the rationale to compare kidney tissue with Human upper tracts tumors.

I would also add a comment in how the absence of blood flow could have changed the results

Author Response

Reviewer 1:

Thank you for your valuable comments, please find attached our responses to the comments you've made. These responses have also been incorporated in the text where applicable according to your suggestions: 

  1. To better specify how tissue block are prepared in a collecting system model

10 fresh non-frozen porcine kidneys were used as ex vivo study model to evaluate laser-tissue ablation. Although various tissue models were used in the literature to investigate the effects of lasers on histological parameters, porcine kidneys were the most reported [20-22]. Kidneys were coronally sectioned and sections of different volumes were obtained from the renal cortex. Only pieces with flat surfaces were selected. A total of 50 tissue blocks of porcine kidney tissue were divided into 5 groups of growing volume sizes tissues of 1, 1.5, 2, 2.5 and 3cm3, forming groups 1, 2, 3, 4 and 5, respectively.

  1. How the Endourological treatment was simulated.

The tissue blocks were placed and fixed in a "renal pelvi-calcyceal" collecting system plastic Model filled by a saline solution (figure 1- D). This model was then accessed with a flexible endoscope and subsequently the tissue was ablated with a laser holmium fiber.

  1. Adding in the methods the rationale to compare kidney tissue with Human upper tracts tumors. 

Thank you for your comment, the following text was added in the methods section: Although various tissue models were used in the literature to investigate the effects of lasers on histological parameters, porcine kidneys were the most reported [15-17,22-23].

Further reference to the use of porcine tissue was also made in the limitations section.

  1. Adding a comment in how the absence of blood flow could have changed the results

Thank you for your comment, the following text was added to highlight the limitations of the study:

This study has some limitations. The first is without doubt the use of porcine kidney tissue, different in its’ histological characteristics than urothelial tumor tissue. However, the conditions in which the experiments were performed, are probably the closest available for the simulation of urothelial tumor ablation, as it is practically impossible to get real tumor tissue on which to run the experiment. Regarding histology, urothelial carcinoma can exhibit a wide range of variant morphologies. Non- invasive neoplasms share a similar morphological spectrum of intra-urothelial changes, ranging from hyperplasia to dysplasia to carcinoma in situ. The main difference from the surrounding non-neoplastic tissues is the architectural growth patterns of the tumor tissue [40]. The second limitation is that the experiments were not performed on live animals with perfused tissues, reproducing the surgical conditions, allowing us to assess hemostatic proprieties [41], and to achieve similar tissue composition, density and temperature compared to in vivo conditions. Therefore, our results may have been affected by this limitation.

Reviewer 2 Report

The present manuscript highlights an important research question. The work has merit and will be of interest to our readers. However, to improve the overall scientific quality of the manuscript, my suggestions are as follows:

Abstract: well-written. No changes are advised.

Introduction: Please elaborate on the findings of the previous studies regarding the different size cutoffs. Also, it is always better to provide a hypothesis at the end of the Introduction section.

Methods: Did you take clearance from the Review Board. Can you provide the application number?

Results: are well presented. No changes are advised.

Discussion: At the end of the discussion section, please elaborate on the limitations.

Also, it is always better to write a paragraph regarding Future proposals and recommendations in the pre-clinical works. How will you ensure safety and efficacy in future clinical studies/trials?

Author Response

Reviewer 2:

Thank you for your valuable comments, please find attached our responses to the comments you've made. These responses have also been incorporated in the text where applicable according to your suggestions:

  • Introduction: Elaboration on the findings of the previous studies regarding the different size cutoffs:

Thank you for your comment, the following text was added :

The mainstay of endoscopic treatment is laser ablation with Holmium:YAG (Ho:YAG) laser, which produces tissue ablation reaching an incision depth (ID) of 2 mm and coagulation thickness of 0.48 mm [15]. Other lasers were also evaluated, and while the Thulium:YAG (Tm:YAG) laser was found to produce larger coagulation areas [16], the new super pulsed Thulium Fiber laser produced a smaller coagulation area and incision depths than the Ho:YAG laser [17].

  • Providing a hypothesis at the end of the Introduction section

Thank you, a reference to this hypothesis was added at the end of the introduction.

  • Methods: Clearance from the Review Board.

The clearance and authorization for the study was in internal institutional process with the accord of the head of department: Professor Olivier Traxer – head of the Urology service in Tenon Hospital APHP Paris and the corresponding author of this paper.

  • Results: are well presented. No changes are advised.

  • Discussion: Elaboration on the limitations.

Thank you for your comment. The following text was added accordingly:

This study has some limitations. The first is without doubt the use of porcine kidney tissue, different in its’ histological characteristics than urothelial tumor tissue. However, the conditions in which the experiments were performed, are probably the closest available for the simulation of urothelial tumor ablation, as it is practically impossible to get real tumor tissue on which to run the experiment. Regarding histology, urothelial carcinoma can exhibit a wide range of variant morphologies. Non- invasive neoplasms share a similar morphological spectrum of intra-urothelial changes, ranging from hyperplasia to dysplasia to carcinoma in situ. The main difference from the surrounding non-neoplastic tissues is the architectural growth patterns of the tumor tissue [40]. The second limitation is that the experiments were not performed on live animals with perfused tissues, reproducing the surgical conditions, allowing us to assess hemostatic proprieties [41], and to achieve similar tissue composition, density and temperature compared to in vivo conditions. Therefore, our results may have been affected by this limitation.

  • A paragraph regarding Future proposals and recommendations in the pre-clinical works.

Thank you for your comment. The following text was added in the limitation paragraph:

This study used Ho:YAG laser only. However, based on previous results of soft tissue ablation with Thulium:YAG and Thulium Fiber Laser (TFL) [16-17,22-23,42], the authors believe that the same results can also be applicable with the use of TFL. Further ex vivo investigations could be performed with TFL.

Reviewer 3 Report

It’s a very interesting and valuable experimental study aiming at defining a maximal volume threshold for UTUC technically amenable for laser ablation using a porcine kidney tissue model. Enjoyed the study design and setup which seems to be well thought-out. 

Some remarks need to be addressed though.

  1. Throughout the text, Authors interchangeably use lesion diameter in cm and lesion volume in cm3. Since EAU Guidelines use a 2cm lesion diameter threshold (see criteria for low risk UTUC - EAU Guidelines), and not the lesion volume threshold, it seems difficult to challenge or reinforce this notion using the models tested in this study. A lesion diameter of 2cm could theoretically translate into a volume of up to 8cm3, assuming a cuboid shape of the tissue block, hence the tissue volumes ablated in the study (up to 3cm3) were actually smaller than what could be encountered in a large proportion of 2cm lesions.
    Nonetheless, the Authors mention (lines 208-209) „…These results carry a significant message regarding the debate of the minimal (?) tissue mass amenable to endoscopic laser treatment. From an endoscopic laser photo-ablative capacity perspective, the minimum (?) limit can be assigned to 3cm lesions….” again having tested a 3cm3 volume block of tissue, the Authors seem to infer „capacity” of laser ablation for 3cm diameter lesions. Please correct where applicable and clarify this issue further in the text.
  2. Secondly, a 2cm diameter threshold mentioned in EAU Guidelines reflects an increasing risk of invasive disease, rather than procedural limitations of laser ablation. Given the difficulties in accurate staging of UTUC with current imaging modalities and biopsy, with significant proportion of pathological upgrading in larger lesions, tumor size served as a surrogate marker for risk stratification. It is obviously debatable and will probably evolve, however this evolution will not driven by technical capacity of laser ablation, rather than by long-term assessment of oncological non-inferiority to RNU. I would afford some space for this in discussion.
  3. As mentioned in discussion, the surgeon experience is key when ablating larger tissue masses. Were the study ablations performed by single surgeon?
  4. In abstract - „study objective”, discussion and conclusions sections Authors refer to a „minimal size of tissue for which total endoscopic ablation with laser is achievable”, whilst in „Introduction” Authors wish to „optimally evaluate, on a histological level, the size limit of lesions which can be managed with laser ablation”. This statements seem contradictory. From the context I gather the Authors should refer to „maximal size” of tissue for which laser ablation is feasible. 
  5. Tables in supplementary files seem duplicates from the main text, no study source data were made available

Author Response

Reviewer 3

Thank you for your valuable comments, please find attached our responses to the comments you've made. These responses have also been incorporated in the text where applicable according to your suggestions:

  1. Throughout the text, Authors interchangeably use lesion diameter in cm and lesion volume in cm3. Since EAU Guidelines use a 2cm lesion diameter threshold (see criteria for low risk UTUC - EAU Guidelines), and not the lesion volume threshold, it seems difficult to challenge or reinforce this notion using the models tested in this study. A lesion diameter of 2cm could theoretically translate into a volume of up to 8cm3, assuming a cuboid shape of the tissue block, hence the tissue volumes ablated in the study (up to 3cm3) were actually smaller than what could be encountered in a large proportion of 2cm lesions.

Thank you for your comment. The following text was added:

Lines 212-214: The results of the current study showed that laser photoablation used in endoscopic treatment of UTUC is capable of entirely ablating a tissue mass as large as 3 cm3, as demonstrated histologically.

Lines 227-230: The authors measured tissue size in cm3 and not cm units, to outline the fact that volume is more indicative of the ability to ablate a mass than its unidimensional size in cm. The importance of using volume in cm3 instead of size has also been highlighted in urinary stones management [38-39].

In urinary stones’ endoscopic treatment, stone burden expressed in volume rather than in largest-diameter has already been demonstrated as a significant determinant of stone-free rate, and accordingly certain publications and guidelines are beginning to consider stone volume instead of stone diameter. We presume that as tumor is also 3D consider its volume should be more precise when discussing the endourological ablation.

  1. the Authors mention (lines 208-209) „…These results carry a significant message regarding the debate of the minimal (?) tissue mass amenable to endoscopic laser treatment. From an endoscopic laser photo-ablative capacity perspective, the minimum (?) limit can be assigned to 3cm lesions….” again having tested a 3cm3 volume block of tissue, the Authors seem to infer „capacity” of laser ablation for 3cm diameter lesions. Please correct where applicable and clarify this issue further in the text. 

Thank you for your comment. We have changed the terms from “minimal” to “maximal” size to avoid and contradiction or misunderstandings. Regarding the results, the intention was –to put it simply- to demonstrate that ablation up to 3cm3 volume is feasible technically with laser energy as proven histologically. We hope that this data can contribute to the discussion about the maximal volume of tumor we can safely ablate in endourological practice.

  1. Secondly, a 2cm diameter threshold mentioned in EAU Guidelines reflects an increasing risk of invasive disease, rather than procedural limitations of laser ablation. Given the difficulties in accurate staging of UTUC with current imaging modalities and biopsy, with significant proportion of pathological upgrading in larger lesions, tumor size served as a surrogate marker for risk stratification. It is obviously debatable and will probably evolve, however this evolution will not driven by technical capacity of laser ablation, rather than by long-term assessment of oncological non-inferiority to RNU. I would afford some space for this in discussion. 

Thank you for this important comment. The text in lines 187-211 addresses specifically this point- the following text was added and followed by references to the works of Pierras, Simone, Villa and Scotland et al, respectively.

Lines 187-193: Tumor size itself is not a component of the TNM classification system in UTUC. However, it is taken into consideration when classifying the tumor into the low or high-risk categories, thus guiding the management choices. Previous recommendations have defined a limit of 1 cm, above which the tumor is considered high risk and endoscopic treatment is not recommended [20-21]. This limit has been changed to 2 cm in the updated recommendations [5]. Nevertheless, as mentioned previously, this limit has been based solely on expert opinions and a few retrospective case series [34-37].

We would like to clear out that the goal in this study was to demonstrate the practical feasibility of laser ablation of different tumor sizes and back it up with histological analysis, not to explore the clinical impact of growing tumor sizes, which is obviously important as you have mentioned.

  1. As mentioned in discussion, the surgeon experience is key when ablating larger tissue masses. Were the study ablations performed by single surgeon? 

The study ablation was performed by two certified fellows in endourology, experienced specialists in urologic surgery in their perspective countries (MC and YB).

  1. In abstract - „study objective”, discussion and conclusions sections Authors refer to a „minimal size of tissue for which total endoscopic ablation with laser is achievable”, whilst in „Introduction” Authors wish to „optimally evaluate, on a histological level, the size limit of lesions which can be managed with laser ablation”. This statements seem contradictory. From the context I gather the Authors should refer to „maximal size” of tissue for which laser ablation is feasible.  

Thank you for this valuable comment. Indeed the goal was the upper limit and therefore we have switched the “minimal” to “maximal”:  “We aim to determinate the maximal size of tissue, for which total endoscopic ablation with laser energy is achievable, from a laser performance perspective”.

  1. Tables :

They were uploaded twice by error, we have removed the supplementary material and added a caption to the table in the manuscript.

Thank you,

Round 2

Reviewer 2 Report

The authors have addressed all my comments in the revised manuscript. The overall scientific quality of the manuscript has increased significantly. The work has merit and will be of interest to our readers. I congratulate the authors for their work.

Reviewer 3 Report

The Authors have addressed my comments sufficiently.